# Migration: A Neglected Potential Contribution of HCl-Oxidized Au(0)

**DOI:** 10.3390/molecules28041600

**Published:** 2023-02-07

**Authors:** Zilong Zhang, Haifeng Zhang, Bolin Wang, Yuxue Yue, Jia Zhao

**Affiliations:** 1School of Chemical Engineering, Northeast Electric Power University, Jilin 132012, China; 2Industrial Catalysis Institute, Zhejiang University of Technology, Hangzhou 310014, China

**Keywords:** Au/C, hydrogen chloride, standard electrode potentials, HCl-oxidized Au, XPS detection

## Abstract

In this study, the typical oxidation process of Au/C catalysts exposed to HCl is presented. Although the process violates the standard electrode potentials, the “oxidized” tendency of Au(0) species is analyzed. This oxidation behavior can only be triggered over the Au/C sample within residual cationic Au species, and terminated over the completely metallic Au(0)/C sample. This study demonstrates that the presence of surface chlorination species cannot facilitate the oxidation of Au(0) and Au(I) when the sample is treated with HCl alone, which excludes the oxidation paths of: Au(0) → Au(III) and Au(I) → Au(III). The reported “HCl-oxidized Au(0)” behavior is partially caused by the migration of Au(III) species in the carbon bulk-phase, which occurs outside the XPS detection limit region and into the detection limit rather than the “HCl-oxidized Au(0)” itself. The mechanism of driving the bulk-phase Au(III) migrated from the steady destabilized state to the carbon surface is then studied. This study demonstrates that the migration of Au cannot be neglected behind the curious oxidation phenomenon by HCl, which provides a new perspective for the oxidation of other noble metals by HCl.

## 1. Introduction

With the rapid development of industrialization, the large amounts of pollutants that can be emitted pose a serious hazard to humans and the environment [1]. Researchers have carried out a wide range of scientific and technical approaches to achieve the full application of clean energy [2]. Catalytic oxidation technology is widely used due to its high efficiency, low operating temperature and lack of secondary pollution [3]. In recent years, precious metal catalysts loaded on activated carbon or “reducible” carriers have been very effective redox catalysts due to their high activity and excellent low-temperature catalytic properties. In previous reports, three-phase catalysts with platinum group metals (PGMs) dispersed as nanoparticles on oxide carriers have been reported to have good application [4]. Nanoscale particle size is used instead of the micron scale because nanoscale leads to an increase in specific surface area, which can lead to large surface energy and an increase in contact area at the reaction interface [5]. Under certain reaction conditions, gold can exhibit higher activity and selectivity than other metal catalysts (including PGMs), which could pave the way for new applications based on equally expensive loaded PGMs for a wide range of applications in the chemical industry. When gold (Au) occurs in the form of nanocrystals, two important observations highlight its unique catalytic behavior: (i) Haruta; low-temperature CO oxidation [6] and (ii) Hutchings; Au is supported on carbon (Au/C) for acetylene hydrochlorination [7], which is a critical industrial process for vinyl chloride monomer production. Assuming that Au will be an effective catalyst for the acetylene hydrochlorination reaction based on a correlation with electron affinity, Au/C has recently been validated as a replacement catalyst for this large-scale industrial process [8]. However, a key advantage of the Au/C catalyst among many noble metals is its good selectivity.

Based on X-ray photoelectron spectroscopy (XPS), it has been demonstrated that hydrogen chloride (HCl) increases the concentrations of the supported Au(III) at the expense of Au(0) [9]. Several experimental studies proved that the metal oxidation process is facilitated by HCl treatment alone [10,11,12,13,14,15]. The active cationic species that have been reduced by C_2_H_2_, forming a metallic state, can be re-oxidized into the oxidation states through the treatment of HCl [16]. The remaining key issue is the nature of “HCl-oxidized Au(0)”. To the best of our knowledge, the zero-point electrode potential of H^+^ from HCl cannot oxidize Au. The underlying mechanism behind this curious phenomenon of “HCl-oxidized Au” has not been clarified yet. Studies [17,18] showed that the presence of large amounts of surface chlorination can facilitate the evolution of Cl_2_ and/or radical (·Cl), combining the activation of the two reactants (C_2_H_2_ + HCl) and triggering the oxidation of Au(I) species. Nevertheless, it is difficult to explain the widely reported oxidation process of Au(0) to Au(III) exposed to HCl alone [19].

In this study, the oxidation process of the Au/C catalysts exposed to HCl is presented, and the “oxidized” tendency of Au(0) species is observed. The nature of Au(0) and Au(III), evolving during the treatment of HCl, is then shown, giving information about the migration behavior of Au(III) species, and providing further information that clarifies the reaction mechanism.

## 2. Results and Discussion

To more intuitively reflect the behavior of “HCl-oxidized Au(0)”, the classical carbon-supported gold (Au/C) catalysts are synthesized with deionized water as impregnation solution, as shown in Appendix A. The reduction of AuCl_3_ is facilitated partly by the reducing nature of the carbon material [20], and partly by the reduced stability of AuCl_3_ in aqueous solutions according to the Au-Cl-H_2_O Pourbaix diagram [17]. Therefore, considerable Au(0) species can be produced with this synthesis strategy. A 1 wt.% Au loading was required to ensure sufficient Au species supplied under the experimental conditions, to avoid possible errors caused by low loading. The metal content was determined by inductively coupled plasma atomic emission spectrometry (ICP-AES) [21] and detected at approximately 0.98 wt.% Au mass, basically matching the expected 1 wt.% Au loading (Table 1). The Au loading was further verified by the X-ray fluorescence (XRF) results, and 1.07 wt.% Au loading was gained. However, when conducting XPS analysis to quantify the supported Au species, a large data gap was presented. Almost 0.01 wt.% was determined, which was approximately 100 times lower than that of the expected 1 wt.% Au loading. To the best of our knowledge, the concentration of the target components in the XPS of samples slightly varied with the repeated tests, showing an error of almost ±20%, which indicates that the quantitative estimations between samples are difficult. However, the contents of Au species that were determined from XPS exceeded this error range. In addition, XPS focuses on surface chemical states [22], which can only detect shallow regions of the sample’s molecular layers [23,24]. This is significantly different from the bulk-phase techniques such as inductively coupled plasma mass spectrometry (ICP-MS), XRF, and temperature-programmed reduction (TPR). In the laboratory instrument used in this study, the inelastic mean free path (IMFP) and the electron take-off angle are almost 1.5 nm and 45°, respectively. Therefore, the information depth is approximately 3.15 nm. In this case, XPS probes are located in the first two molecular layers. The difference of 100 times indicates that most of the supported Au species are located in the bulk-phase of the carbon host, rather than the 3 nm region in the surface-phase. Similar spatial distribution status of Au species can be confirmed from the XPS results presented in previous studies [9,23].

Afterwards, the evolution behavior of the supported Au/C catalysts with HCl sequence pretreatment was studied. The changes in the C1s and O1s spectra were too small to be detected, as shown in Figure 1a,b. However, small differences in the Cl 2p spectra existed. To the best of our knowledge, the XPS binding energies of Cl 2p centered at the low (<198 eV, Figure 1c) and high (>200 eV) levels can be assigned to the inorganic Cl^-^ (Au-Cl species) and organic Cl^-^ (C-Cl species), respectively. The presented Au-Cl and C-Cl species over the fresh Au/C catalyst before pretreating with HCl were rationalized as a result of the preparation of catalysts. Hutchings et al. [17] showed that the aqueous impregnation of HAuCl_4_ on the carbon support consists of the deposition of metallic Au(0), and the presumably consequent formation of a greater amount of C-Cl and few Au-Cl species [17]. This observation was consistent with the assignments of XPS binding energies, since large amounts of Au(III) were generated over the fresh Au/C catalysts, as marked with red arrow in Figure 1d. When HCl started to treat Au/C, the Au 4f and Cl 2p spectra showed a clear accompanying change trend and a significant increase with the treating time, while there was little effect on the C 1s and O 1s spectra as shown in Figure 1a,b. When the oxidation contribution of inorganic Cl^-^ species was excluded [17], the oxidation process from Au(0) to Au(III)/Au(I) seemed to be facilitated by the organic C-Cl species, as marked with red arrow in Figure 1c. Previous studies have proved that the exposure to HCl alone does not result in the oxidation of Au(I) to Au(III) species [17]. In addition, the observed induction period related to the oxidation of Au(I) to Au(III) can only be generated when the sample is simultaneously treated with both reactants (C_2_H_2_ + HCl) [17]. Therefore, in the experiment presented in this paper, the growth of Au(III) species was more likely to have originated from the “oxidized Au(0)” with the treatment of HCl alone, which was the typical origin phenomenon of “HCl oxidized Au(0)” reported in this field [9,10,11,12,13,14,15,19]. If the reduction of Au(III) by X-rays is considered [20], more Au(III) will be produced during HCl treatment, which makes the oxidation behavior more conclusive. 

Previous studies deduced that the presence of large amounts of surface C-Cl species can facilitate the evolution of Cl_2_ from the surface within both reactants (C_2_H_2_ + HCl), which is a strong enough oxidant to convert Au(I) into Au(III) in addition to the residual surface NO_x_ species [17]. The results obtained in this study confirmed the rationality of this speculation, since the peak intensity of the C-Cl species sharply decreased when C_2_H_2_ was pulsed into the HCl stream, as shown in Appendix A. On the contrary, the surface C-Cl species were facilitated when individual HCl was introduced (Figure 1c), which can be attributed to the chlorination of HCl on carbon [25]. As a result, it can be concluded that C_2_H_2_ is essential for the evolution of Cl_2_ from the surface C-Cl species. Moreover, the strong Cl_2_ oxidant cannot be released to oxidize Au, regardless of whether Au is in state (I) or (III), without the assistance of C_2_H_2_. The phenomenon of “HCl-oxidized Au(0)” should be further explored.

It was deduced that the aforementioned “oxidation” behavior disappeared when the Au(III) was quantified using the bulk-phase TPR technique. The Au(III) content slightly changed from 9.65 and 9.12 to 10.08 μmol g(Cat.)^−1^ within the HCl treatment (Table 1), which contradicts the XPS results. To more intuitively compare the changes in Au(III), the relative Au(III)/Au(0) ratio was calculated. It was deduced that the behavior of “HCl-oxidized Au” was significant in the result based on XPS (0% vs. 56%, Table 1). However, it disappeared from the TPR result (19.01% vs. 18.60%). The Au(III)/Au(0) ratio ranged between 19.01% and 18.60% within 60 min of HCl treatment, which was likely to have been caused by experimental errors, because it was difficult to ensure that all the Au species were completely consistent in any area of the carbon host. In addition, if the phenomenon of Au(0) being oxidized by HCl is as clear as the XPS results, the value of Au(III)/Au(0) in the TPR results should be rather high, since 60 min of HCl treatment was sufficient for oxidizing Au(0) wherever it was located, in the surface-phase or bulk-phase. In order to visualize the evolution of the chemical state of Au species more closely, a control experiment was carried out in which all cationic gold species over Au/C catalysts were reduced to the (0) valence state by H_2_. It can be clearly seen from the last row of Table 1 that no Au(0) was oxidized to Au(III), either detected from the surface-phase or bulk-phase technique, confirming that Au(0) cannot be oxidized by HCl alone, which seems to be corroborated by the rules based on the standard electrode potential.

Consequently, it is assumed that the significantly increased Au(III) in the region of approximately 3 nm within the XPS detection limit was probably induced by the migration of Au(III) in the bulk-phase (>3 nm depth) to the surface-phase during the HCl treatment process, rather than the widely reported “HCl-oxidized Au”.

To verify this hypothesis, three questions should be answered: How is Au(III) (i) stabilized and (ii) destabilized? (iii) What is the driving force for the migration of Au(III) species from the bulk-phase to the surface-phase?

To the best of our knowledge, many functional groups are presented on the carbon host. The acidic groups that result in a polar and hydrophilic nature of the carbon surface are usually responsible for the adsorption of metal ions from aqueous solutions, especially for the Au/C system [26]. Therefore, the functional groups were analyzed using the Boehm method. The average amounts of the groups detected on the carbon surface are presented in Appendix A. The lactonic groups represented strong acidic groups, while the phenolic and carbonyl groups were weakly acidic groups [27]. However, the content of lactonic groups was too low to study their stabilization effect on Au(III). As a result, the absolute amount of advantageous phenolic groups (Appendix A) was considered in order to investigate their stabilizing effect on Au(III). To rule out the effect of functional group changes during HCl treatment, the distribution of functional groups was also determined under this condition. It can be seen from Appendix A that the absolute amount of phenolic groups changed little. For the Au/C catalyst, the dominance of phenolic groups can be observed and clarified from their corresponding O1s spectrum (>532 eV, Figure 1b) [28].

The CO temperature-programmed desorption–mass spectrometer (TPD-MS) technique was used to study the interaction between the Au(III) species and function groups. Four desorption peaks in the range of 518–850°C were observed for the evaluated samples (Appendix A), which can be assigned to carboxylic anhydride, phenol, ether, and carbonyl [27,29]. It can be clearly seen that the phenolic groups, centered at almost 700 °C, govern the dominant groups, which is consistent with the Boehm result. The amount of CO desorption belonging to the phenolic groups was significantly reduced, from 217 μmol g(Cat.)^−1^ to 155 μmol g(Cat.)^−1^ when Au was supported on carbon, as shown in Appendix A. Previous studies [29,30] demonstrated that the metal deposition may contribute to the significant decrease in the surface oxygen released upon decomposition or surface reconstruction. Therefore, strong metal–support interaction was presented between the Au(III) and phenolic groups. In addition, Au(III) was more likely to be bonded with phenolic groups (Equation (1) in Figure 2). Bulushev et al. [30] proved that the phenolic groups interact with the Au(III) precursor by anchoring the cationic Au species. In the subsequent HCl treatment, the amount of CO desorption significantly decreased from 155 μmol g(Cat.)^−1^ to 36 μmol g(Cat.)^−1^, which indicated that this strong interaction was unstable, destroying the -Au-O-C- bonds and then generating the -C-Cl- bonds (Equation (2) in Figure 2). The detected H_2_O signal (Appendix A) and C-Cl bonds (Appendix A) for the in situ HCl-treated Au/C confirmed this speculation.

The free energy of the Au species was then calculated, as shown in Figure 2. The migration process started with the adsorption of AuCl_3_ over phenolic -OH groups, which results in the dissociation of -OH species followed by the formation of HAuCl_3_-O * species with a reaction energy of −0.52 eV. The exposure of the HAuCl_3_-O * species to free HCl molecules resulted in their further renormalization. It was observed that the Cl atom from HCl attacks the O atom of the HAuCl_3_-O * species, forming •OH radical. In addition, a new bond is formed between the H atom of HAuCl_3_-O * and the •OH radical, leading to the formation of a water molecule and AuCl_4_ * species. Furthermore, excessive HCl attacked AuCl4 *, resulting in the formation of free HAuCl4 molecules that escaped into the gas phase from the support. With only HCl involved in the migration process, the first step requires crossing over a 1.96 eV/mol energy barrier. The energy barrier for the second step was 0.44 eV/mol, which was large enough to make the destabilized Au migrate to the carbon surface located in the XPS detection limit region, resulting in the phenomenon of “HCl-oxidized Au”.

## 3. Conclusions

This study presented the oxidation process of Au/C catalysts exposed to HCl, and analyzed the “oxidized” tendency of Au(0) species. The obtained results showed that the oxidization of Au(0) by HCl is caused by the migration of Au(III) in the bulk-phase to the surface-phase region detected in XPS. Although it is difficult to observe the migration of Au species in view of the corrosion of HCl to the directly in situ instrument, this study provides a new perspective on the oxidation process of metals by HCl, especially for the oxidation process that is difficult to trigger based on the standard electrode potential. More importantly, this study further explores and enriches the dynamic evolutionary behavior of noble metal catalysts under oxidizing atmospheres, providing a reference for the development of low-cost, high-performance catalysts for vinyl chloride monomer (VCM) synthesis.

## Figures and Tables

**Figure 1 molecules-28-01600-f001:**
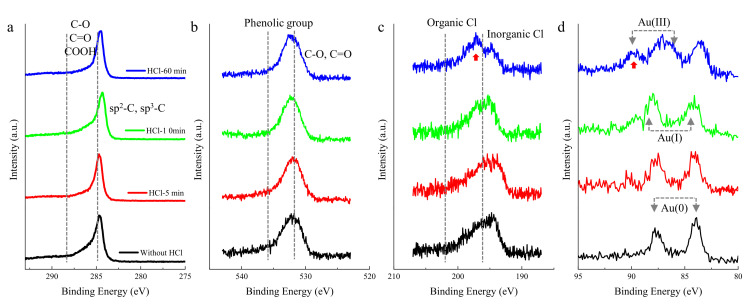
XPS spectrum of (**a**) C 1s, (**b**) O 1s, (**c**) Cl 2p, and (**d**) Au 4f XPS spectra of Au/C catalysts with HCl sequence pretreatment.

**Figure 2 molecules-28-01600-f002:**
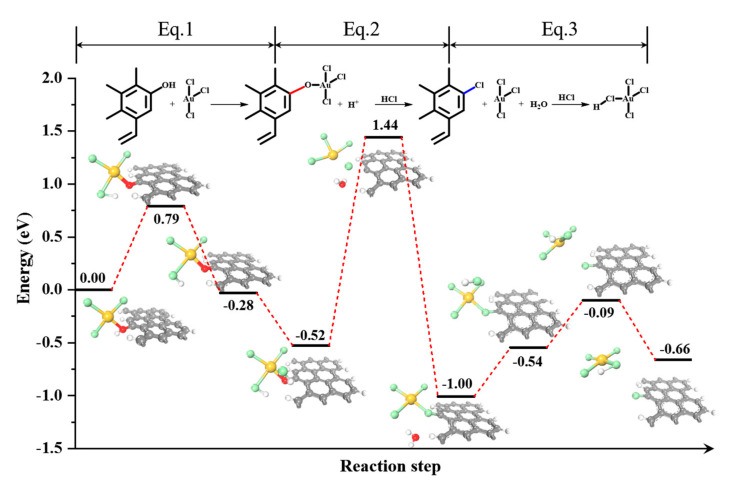
Reaction energy profile for proposed Au migration process. The green, gray, red, white, and yellow balls represent Cl, C, O, H, and Au atoms, respectively.

**Table 1 molecules-28-01600-t001:** Surface and bulk composition of Au/C catalyst.

Catalysts	Composition							
	Au/C (wt.%)				Au(III) (μmol g(Cat.)^−1^)		Au(III)/Au(0) (mol.%)	
	Nominal	ICP	XPS ^a^	XRF	XPS ^a^	TPR ^b^	XPS ^a^	TPR ^b^
Au/C	1.00	0.98	0.01	1.07	0	9.65	0	19.01
Au/C HCl-5 min	1.00	1.01	0.04	0.95	0.005	9.12	23	17.97
Au/C HCl-10 min	1.00	0.98	0.08	1.02	0.020	10.08	51	19.86
Au/C HCl-60 min	1.00	0.97	0.16	0.98	0.045	9.44	56	18.60
Au(0)/C H_2_-180 min, HCl-60 min	1.00	1.03	0.01	1.11	0	0	0	0

^a^ Determined from the XPS analysis in Appendix A; ^b^ determined from the H_2_-TPR analysis in Appendix A.

## Data Availability

The data presented in this study are available in the article itself.

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
