# Peer review of "Migration: A Neglected Potential Contribution of HCl-Oxidized Au(0)"

_molecules, 2023, doi:10.3390/molecules28041600_

Round 1
Reviewer 1 Report
This study repeated the oxidation process of Au/C catalysts exposed to HCl, and observed the “oxidized” tendency of Au(0) species. This study provides a new perspective on the oxidation process of metals by HCl. I recommend it published after the following revisions and questions addressed.
1. Some of the basic errors in the references are due to carelessness, such as: 7, 9, and 19.
2. For inductively coupled plasma atomic emission spectrometry (ICP-AES), X-Ray-Fluorescence (XRF), and X-ray photoelectron spectra (XPS), etc. in the introduction, I suggest just writing the abbreviation in the follow-up.
3. The Figure S2 is important and should be transferred from supporting information to the manuscript.
4. Whether the results of lines 206-210 are based on evidence or corresponding references are provided.
5. Some important papers should be cited "Angew. Chem. Int. Ed.,DOI: 10.1002/anie.202209350;Advanced Materials, 2021, 33, 2105163;Advanced Materials, 2022, 34, 2107836".
Reviewer 2 Report
The authors describe the oxidization of Au(0) by HCl as caused by the migration of Au(III) in the bulk phase to the surface phase region. This is the major innovation of the work. The experimental and characterization results look very impressive. However, there are not enough Figures in the main article to help potential readers, which makes the article difficult to understand. Moreover, the paper's grammar and academic expression are rough. It is recommended to further improve and polish the language, otherwise, it will be difficult to attract readers, and it will be difficult for readers to obtain effective information from the article. Another point is that It is unreasonable that the summary section and abstract section have almost identical sentences. Although the figure and language need further beautification, as proof-of-concept the work is novel. I recommend publishing as is.
Reviewer 3 Report
In this study, the typical oxidation process of Au/C catalysts exposed to HCl is repeated, and the ‘oxidized’ tendency of Au(0) species is observed, although the process violates the Standard Electrode Potentials. The mechanism of driving the bulk-phase Au(III) migrated from the steady-destabilized-state to the carbon surface is studied. This study reveals that the migration of Au cannot be neglected behind the curious oxidation phenomenon by HCl, which provides a new perspective for the oxidation of other noble metals by HCl. However, before considering publication, there are some issues that need to be clarified. The detailed comments are as follows:
1. The graphical abstract needs to be added.
2. some noble catalysts should be introduced in the introduction. Why select Au. Some noble catalysts references should be added. Such as Environmental Science & Technology, 2022, 56(23) 17321-17330; Environmental Functional Materials 1 (2022) 166-181; Journal of Catalysis, 2023, 418, 90-99; Journal of Catalysis, 2022, 413, 31-47.
3. Please describe the advantages of Au/C catalysts and their prospects for large-scale industrial applications in detail.
4. Line no.110, (Fig. S2a-b) should be changed to (Figs. S2a-b).
5. Au is a precious metal, and whether its high price will hinder the industrialization of Au/C catalysts?
6. Is it necessary to draw the material synthesis steps?
7. In the conclusion, please describe the significance of this study in more detail.
8. Some tense problems, grammatical problems and the beauty of pictures in the article need to be further improve.
Round 2
Reviewer 3 Report
accepted